# Effect of Sprouted Buckwheat on Glycemic Index and Quality of Reconstituted Rice

**DOI:** 10.3390/foods13081148

**Published:** 2024-04-10

**Authors:** Lingtao Kang, Jiaqian Luo, Zhipeng Su, Liling Zhou, Qiutao Xie, Gaoyang Li

**Affiliations:** 1Longping Branch, College of Biology, Hunan University, Changsha 410125, China; klt0427@hnu.edu.cn (L.K.); luojq199909@163.com (J.L.); suzp15@163.com (Z.S.); zhoull203@163.com (L.Z.); 2Hunan Agricultural Product Processing Institute, Hunan Academy of Agricultural Sciences, Changsha 410125, China; jiningxqt@126.com; 3Hunan Provincial Key Laboratory for Fruits and Vegetables Storage Processing and Quality Safety, Changsha 410125, China

**Keywords:** sprouted buckwheat, reconstituted rice, glycemic index

## Abstract

This study utilized sprouted buckwheat as the main component and aimed to optimize its combination with other grains to produce reconstituted rice with enhanced taste and a reduced glycemic index (GI). The optimal blend comprised wheat flour, sprouted buckwheat flour, black rice flour, and purple potato flour in a ratio of 34.5:28.8:26.7:10.0. Based on this blend, the reconstituted rice processed through extrusion puffing exhibited a purple-black hue; meanwhile, the instant reconstituted rice, produced through further microwave puffing, displayed a reddish-brown color. both imparted a rich cereal flavor. The starch in both types of rice exhibited a V-shaped structure with lower relative crystallinity. Compared to commercial rice, the reconstituted rice and instant reconstituted rice contained higher levels of flavonoids, polyphenols, and other flavor compounds, along with 1.63-fold and 1.75-fold more proteins, respectively. The GI values of the reconstituted rice and the instant reconstituted rice were 68.86 and 69.47, respectively; thus, they are medium-GI foods that can alleviate the increase in blood glucose levels.

## 1. Introduction

Buckwheat is one among the group of important homologous crops with medicinal and food-related value [1]. Abundant in nutrients, it comprises starch, proteins, amino acids (including the eight essential amino acids), unsaturated fatty acids (like linoleic acid, oleic acid, etc.), vitamins, trace elements (such as potassium, calcium, magnesium, etc.), and dietary fiber [2]. Additionally, it contains flavonoids, polyphenols, and other functional compounds [2,3]. Various flavonoids like rutin, quercetin, and kaempferol, along with phenolic compounds such as gallic acid, protocatechuic acid, and p-hydroxybenzoic acid, are found in buckwheat. These functional components are associated with antioxidant properties [4], regulation of blood glucose and blood lipid levels [5], anticancer properties [6], and alleviation of alcoholic liver injury [7]. Because of its high nutritional content and functional properties, buckwheat has received increasing interest from both consumers and food researchers. However, due to its apparent bitter taste and poor palatability, there has been limited use of buckwheat in functional foods [8].

Sprouting is an effective method of enhancing the edible value of grains and coarse cereals, and it can improve the culinary characteristics and sensory quality of foods [9]. A series of complex physiological and biochemical reactions occur during seed sprouting, wherein the stored nutrients such as starch and proteins are degraded [10], γ-aminobutyric acid and phenolic compounds are synthesized [11,12], antinutritional factors such as phytic acid are reduced [13], and the nutritional value of the grain and coarse cereal is increased to a certain extent, with improvement in taste. Some studies have reported on the nutritional value and functional properties of sprouted buckwheat. For example, Zhou Xiaoli et al. showed that, after 7 days of buckwheat germination, vitamin C content increased to 14-fold compared with the original content; the content of total flavonoids increased to 4.2-fold compared with the original content; rutin content increased by approximately 300%; and the scavenging activity of DPPH, ABTS, and superoxide radicals increased by 107%, 144%, and 88%, respectively [14]. Seerat Bhinder et al. studied the effect of different sprouting time periods (24–96 h) on the nutrient composition of buckwheat and the production of buckwheat muffins; the authors found that sprouting for more than 72 h enhanced the antioxidant activity of buckwheat flour (due to the accumulation of free flavonoids and phenolic acids); sprouting for 48–72 h increased the content of essential amino acids and γ-amino butyric acid in buckwheat flour; and the addition of buckwheat flour with 24–48 h of sprouting to muffins led to a lower glycemic index (GI) than that observed in muffins made from ungerminated buckwheat flour [2].

Extruded reconstituted rice, also known as compound rice, artificial rice, or nutritional composite rice, is obtained from a variety of crops and vegetables as raw materials, and it is processed into food products through extrusion, ripening, drying, and other processes; the appearance and consumption characteristics of this rice are similar to those of ordinary rice [15]. Reconstituted rice has a high degree of pasting and a short cooking time. This not only helps in minimizing nutrient loss from the raw materials but also enhances the nutritional value of reconstituted rice as compared to natural rice [16]. Additionally, it lowers energy consumption, thereby rendering it a convenient, efficient, environmentally friendly, and energy-saving option. In our prior research, it was observed that sprouting buckwheat and subjecting it to heat treatment (by boiling for 1 min) led to a general augmentation in active substances, resulting in enhanced antioxidant capacity and an α-amylase inhibition effect. For more efficient use of buckwheat in the functional components and for simultaneously improving the bitter taste, poor palatability, and other limitations of buckwheat flour, in the present study, 4-day-sprouted, heat-treated buckwheat was used as the core raw material. This was compounded with wheat flour, black rice flour, and purple sweet potato flour through extrusion to prepare reconstituted rice. Currently, few studies have reported on preparing reconstituted rice with sprouted buckwheat flour after compounding in terms of changes in textural properties, nutrient changes, and the ability of reconstituted rice to alleviate the increase in blood glucose levels. This study provides theoretical data to support the development of reconstituted rice containing sprouted buckwheat for alleviating the increase in blood glucose levels for further guidance regarding the application of reconstituted rice.

## 2. Materials and Methods

### 2.1. Materials and Reagents

The buckwheat variety used was Jin buckwheat No. 2. Sprouted buckwheat flour was prepared in the laboratory. Wheat flour, black rice flour, purple sweet potato flour, rice, and instant rice were purchased from the market. Starch glucosidase (enzyme activity ≥ 100 U/mg) and α-amylase (enzyme activity ≥ 4000 U/g) were purchased from Shanghai Ruiyong Biotechnology Co., Ltd. (Shanghai, China). Thermostable α-amylase (10,000 U/g) was purchased from Xingtai Wanda Bioengineering Co., Ltd. (Xingtai, China). All other chemical reagents were of analytically pure grade and were purchased from the domestic market.

### 2.2. Preparation of Sprouted Buckwheat Flour

Buckwheat seeds were washed and soaked at room temperature for 4 h. Subsequently, the seeds were evenly dispersed on the germination tray and incubated in the dark at 25 °C for 4 days. The sprouted buckwheat seeds were placed in boiling water and heated for 1 min, removed, and then cooled to room temperature; the sprouted seeds were subjected to constant temperature drying at 40 °C for 4 h, hulled, and crushed through a 40-mesh sieve. The crushed seeds were collected in a sealed bag and stored at −80 °C.

### 2.3. Formulation Design and Optimization of Sprouted Buckwheat Reconstituted Rice

#### 2.3.1. Mixing Test Design

Mixing experiments were designed by setting constraints using Design-Expert 8.0.6 software. A pre-test revealed that the sugar content was high because of the viscosity of purple sweet potato (D). A high sugar content could lead to difficulties in product formation; hence, the sugar content percentage was fixed at 10%. The percentages of wheat flour (A) and sprouted buckwheat flour (B) content were set in the range of 20% ≤ A ≤ 50%. The range of variation in the black rice flour (C) content percentage was set to 10% ≤ C ≤ 30%. The combination of A+B+C was 90%; thus, 16 recipes were yielded. In this experiment, the raw material percentage was used as the independent variable, and the α-amylase inhibitory activity, resistant starch content, and sensory score of the product were used as the response values.

#### 2.3.2. Extrusion Puffing Process and Parameters

Extrusion process: Cereal raw materials were ground in a 40-mesh sieve, and their moisture level was adjusted. Subsequently, extrusion puffing was performed to yield reconstituted rice, and microwave puffing was carried out to obtain instant reconstituted rice. The obtained rice was dried, refrigerated, and preserved.

Extrusion puffing parameters: The raw materials were crushed and sieved, and the moisture content was adjusted to 22%. The raw materials were extruded and expanded using a twin-screw extruder (LY70, Linyang Machinery, Jinan, China) with the following parameters: screw speed: 20.1 Hz; temperature: 40.5 °C in zone 1, 62.6 °C in zone 2, and 68.7 °C in zone 3.

In microwave puffing, expansion was performed for 1 min.

#### 2.3.3. Determination of α-Amylase Inhibition Rate and Resistant Starch Content and Sensory Evaluation

The α-amylase inhibition rate was determined by referring to the method reported by Peng Xi et al., with minor modifications [17]. Briefly, 5 g of the sample was taken, and 0.1 mol/L PBS (pH 6.9) was added and shaken well; the sample was centrifuged at 3000 rpm for 10 min, and the supernatant was measured. Next, 500 µL of the supernatant was taken, and 500 µL of the α-amylase solution was added and incubated at 37 °C for 10 min. Next, 500 µL of 1% soluble starch was added and incubated at 37 °C for 10 min. Subsequently, 1 mL of dinitrosalicylic acid (DNS) reagent was added, and the reaction was terminated in a boiling water bath for 5 min. Following cooling to room temperature and one-fold dilution, the absorbance at 540 nm was determined using a microplate reader (Synergy H1, Biotek, USA). In the negative group, 500 µL of PBS replaced the supernatant. For the blank group, 1 mL PBS replaced both the supernatant and α-amylase solution. The inhibition rate of α-amylase was computed using the following formula: inhibition rate (%) = (A1 − A2) − (A3 − A2)/(A1 − A2) × 100%, where A1 represents the negative group, A2 denotes the blank group, and A3 signifies the sample group.

Resistant starch content was determined according to the method of Goni et al., with minor modifications [18]. Briefly, 200 mg of sample was weighed in a 50 mL centrifuge tube and digested successively by pepsin and cellulase; the pH of the solution was adjusted to 6.0, and 1 mL thermostable α-amylase (20 mg/mL) was added and shaken at 90 °C for 60 min. The pH of the solution was adjusted to 4.5 after cooling to room temperature. Next, 1 mL of amyloglucosidase (20 mg/mL) was added, shaken at 60 °C for 60 min, and then centrifuged for 5 min at 8000 rpm; the supernatant was discarded, and the steps of washing with water and centrifugation were repeated three times. Next, 2 mL of KOH (2 mol/L) was added to the precipitate and shaken at room temperature for 30 min. The pH was adjusted to 4.5, and 1 mL of amyloglucosidase (20 mg/mL) was added. The solution was shaken at 60 °C for 60 min in a water bath. After cooling, the solution was centrifuged at 8000 rpm for 5 min. The supernatant was added to a 100 mL volumetric flask. The precipitate was subjected to washing with distilled water 3 times, centrifuged, and then added to a volumetric flask; the volume of the solution was adjusted to 100 mL with distilled water, and the solution was shaken well and prepared for use. The glucose content was determined by the DNS method and then converted to the amount of resistant starch. The resistant starch yield was calculated according to the following formula: resistant starch yield (%) = (m1 × 0.9 × 100)/m2. Here, m1 is the mass of glucose (mg); m2 is the mass of the sample (mg); and 0.9 is the conversion coefficient between glucose and starch.

Sensory evaluation of the samples was performed by a professionally trained 10-member tasting panel for the following four aspects: morphology (25 points), color (20 points), taste (30 points), and flavor (25 points). The specific scoring criteria are shown in Appendix A.

### 2.4. Nutrient Content Determination

The contents of starch, fats, proteins, moisture, and ash in the sample were determined according to the following standard methods: GB5009.9-2016, GB/T5009.6-2003, GB/T5009.5-2010, GB5009.3-2016, and GB/T5009.4-2016, respectively. The water solubility index was determined by referring to the method of Junling Wu et al. [19].

### 2.5. Determination of Total Flavonoids and Phenols

Extract preparation: The sample was combined with a certain volume of 70% methanol solution and sonicated for 40 min. After completing three rounds of centrifugation, 70% methanol solution was added to the precipitate, and the supernatant was combined in a 50 mL brown volumetric flask.

The method of Seerat Bhinder et al. was adopted to determine the total flavonoid and total phenol content, with slight modifications [2]. The total flavonoid content was determined as follows: (i) 50 µL of sample extract was aspirated, and 20 µL of 5% NaNO_2_ solution was added and allowed to stand for 5 min; (ii) 20 µL of 10% AlCl_3_ solution was added and allowed to stand for 5 min; and (iii) finally, 200 µL of 4% NaOH solution was added and allowed to stand for 10 min. The absorbance was measured at 510 nm by using a microplate reader (synergy H1). The total flavonoid content (%) was determined as follows: C × V/W. Here, C is the mass concentration of flavonoids calculated from the standard curve equation, mg/mL; W is the mass of the sample, g; and V is the total volume of the reaction system, mL. The total phenol content was determined as follows: (i) the extraction solution was mixed with the Folin-phenol reagent in a 1:1 ratio, and 1.5 mL of Na_2_CO_3_ solution was added; (ii) water was added to achieve constant volume, mixed well, and the reaction mixture was kept in the dark for 2 h; (iii) absorbance was measured at 765 nm using a UV–Vis spectrophotometer (UV-1800, Shimadzu Instruments Co., Ltd., Suzhou, China). The total phenol content (%) was estimated as C × V/W, where C is the mass concentration of total phenol calculated from the standard curve equation, mg/mL; W is the mass of the sample, g; and V is the total volume of the reaction system, mL.

### 2.6. In Vitro Digestive Characterization

The method of Seerat Bhinder et al. was used, with appropriate modifications [2]. Briefly, 500 mg of sample was weighed, and 10 mL of 0.2 mol/L sodium acetate buffer was added and boiled for 30 min. After cooling to room temperature, the sample was placed in a water bath at 37 °C for 5 min, followed by the addition of 1 mL of α-amylase and 4 mL of amyloglucosidase. Samples were collected at 0, 10, 30, 60, 90, 120, 150, and 180 min for 0.5 mL, followed by the addition of 4.5 mL of anhydrous ethanol. Following centrifugation at 4000 rpm for 15 min, the supernatant was removed, and the glucose content of the sample was assessed using the DNS method to determine the rate of starch hydrolysis in the sample. The starch hydrolysis rate was computed using the following formula: G_t_ × 0.9/m × 100%. Here, G_t_ represents the amount of glucose produced at time t of hydrolysis, 0.9 denotes the conversion factor between glucose and starch, and m indicates the sample mass. The hydrolysis index (HI) was determined as follows: A_0_/A × 100%. Here, A_0_ signifies the area under the hydrolysis curve of carbohydrates available for various samples within 0 to 180 min, and A denotes the area under the hydrolysis curve of carbohydrates available for the reference sample (white bread) during the same period. The formula for calculating the GI was as follows: 39.71 + 0.549HI.

### 2.7. Colorimetry and X-ray Diffraction

Colorimetry: The samples were placed in a cuvette and the *L**, *a**, and *b** values were determined with a colorimeter (CR-400, Minolta, Japan). The total color difference Δ*E* was calculated as follows: $\Delta E=\sqrt{{\left(L^{*}-L_{S}^{*}\right)}^{2}+{\left(a^{*}-a_{S}^{*}\right)}^{2}+{\left(b^{*}-b_{S}^{*}\right)}^{2}}$. Here, *L_s_**, *a_s_**, and *b_s_** are the measured values of the standard white plate.

X-ray diffraction (XRD): An XRD instrument (Smartlab 9 kW, Rigaku, Japan) was used to measure XRD under the following parameters: tube voltage, 40 kV; current, 40 mA; scanning range, 5° to 60°; and scanning speed, 2°/min. The crystallinity of the samples was analyzed and quantified using Origin2019 software. The RC (%) was determined as follows: (F_t_ − F_n_)/F_t_. Here, F_n_ is the non-crystalline area, and F_t_ is the total area of the diffractogram.

### 2.8. Textural Properties and Scanning Electron Microscopy

Textural properties: The rice grains were first steamed and cooled, and full grains of uniform size were selected for measurement. An instrument (CT3, Brookfield, MA, USA) was used with the following parameters: test probe, TA9; test speed, 2 mm/s; compression interval, 5 s; trigger value, 10 g; and compression deformation, 60%.

Scanning electron microscopy (SEM): The sample was fixed on a carrier stage for gold spraying, and the microstructure of the sample powder was observed using a scanning electron microscope (SU8000, Hitachi, Tokyo, Japan) with a magnification of ×2000.

### 2.9. Thermal Properties and Pasting Properties

Digital scanning calorimetry (DSC): By using an instrument (DSC 214 Polyma, NETZSCH, Heidelberg, Germany), certain amounts of the sample and distilled water were weighed, added to a crucible, and sealed, and a blank disk was used for comparison. The starting temperature was 10 °C, and the termination temperature was 130 °C, with a temperature increase rate of 10 °C/min.

Pasting properties: Pasting properties were determined using a rapid viscosity analyzer (model 4500, Perten Instruments, Hägersten, Sweden). The appropriate amount of sample was weighed into a special aluminum box, and 25 mL of distilled water was added. The mixture was stirred for 10 s at 960 rpm and then maintained at 160 rpm until the end of the experiment. Other measurement conditions were referred based on GB/T 24852-2010.

### 2.10. Determination of Volatile Components

Volatile components were determined by a gas chromatography-mass spectrometry (GC-MS 7890A-5975C, Agilent, CA, USA) by referring to the method of Yan Li et al. with minor modifications [20].

Headspace solid-phase microextraction: Briefly, 1 g of sample powder, 9 mL of distilled water, and a small magnetic stirrer were placed in a 20 mL headspace extraction vial. Next, 20 µL of the internal standard cyclohexanone (50 mg/L) was added, rapidly sealed, and inserted into a manual injector fitted with a 50/30 µm DVB/CAR/PDMS fiber head. The mixture was then magnetically stirred and subjected to headspace extraction for 1 h at 100 °C. The extraction head was then slowly removed and inserted into the inlet port.

Chromatographic column: The J&W DB-5 quartz capillary column (30 m × 0.25 mm, 0.25 µm) was used. The temperature increase program was as follows: initial column temperature, 40 °C; increase in temperature to 280 °C at 5 °C/min; and holding at this temperature for 10 min. The total run time was 58 min. The temperature of the vaporization chamber was 250 °C. The carrier gas was high-purity helium (He), with a flow rate of 1.0 mL/min and pressure of 2.4 kPa. The injection volume was 1 µL, and the solvent delay time was 1 min.

Identification of volatile substances: By using the software available in the mass spectrometry unit, the peaks appearing in the total ion flow diagram were systematically searched, and the spectra were checked. The concentration of each chemical component was calculated using the internal standard method according to the following formula: Cx=n×Cis. Here, *C_x_* is the concentration of compound X, µg/mL; n is the peak area ratio of compound X to the internal standard; and C*_is_* is the concentration of the internal standard in the sample, µg/mL.

### 2.11. Statistical Analysis of Data

Design-Expert was used to design and optimize the mixing experiments. ANOVA in SPSS Statistics version 23 was used to analyze significant differences in the data. Origin 2019 and Word 2016 were used to plot the graphs.

## 3. Results and Discussion

### 3.1. Weighted Score Response Value Analysis and Determination of Optimal Formulation

The analytical results of α-amylase inhibition, resistant starch content, and sensory scores as response values are provided in the Appendix A. The following regression equation was constructed through multiple regression fitting of the experimental data for the weighted scores shown in Table 1: Y_1_ = 45.59A + 67.70B − 265.2C − 80.75AB + 611.88AC + 551.99BC − 545.25ABC + 82.98AB (A − B) − 488.68AC (A − C) − 456.54BC (B − C). The results of the weighted score regression and ANOVA are shown in Table 2. The regression model was significant (*p* < 0.05), the lack of fit was not significant, and the model was well fitted to the test results. The regression equation Y_1_ showed that the response coefficients of A, B, AC, and BC were positive; thus, these results indicate that the use of only wheat flour, sprouted buckwheat flour, a wheat flour-black rice flour combination, and a sprouted buckwheat flour-black rice flour combination had a promotional effect on the weighted scores in the following order: sprouted buckwheat flour > wheat flour and wheat flour-black rice flour combination > sprouted buckwheat flour-black rice flour combination. This finding showed that wheat flour and sprouted buckwheat flour played an important role, and the synergistic effect of wheat flour, black rice flour, and the sprouted buckwheat flour-black rice flour combination significantly increased the weighted scores (*p* < 0.01). Figure 1 shows the contour plot of the effect of compound ingredient ratios on the weighted scores.

Sensory and functional characteristics, two important attributes of functional foods, should be considered in an integrated manner during food development. Based on the experimental design and results of the analysis, the optimal formulation of reconstituted rice was wheat flour/sprouted buckwheat flour/black rice flour/purple sweet potato flour in a ratio of 34.5:28.8:26.7:10.0. The predicted value of the weighted score was 52.70. For this optimized formulation, the validation values of α-amylase inhibition, resistant starch content, sensory score, and weighted score were (51.63 ± 3.82)%, (42.11 ± 0.94)%, 75.78 ± 4.82 and 52.65, respectively. The validation test results confirmed that the model predictions were consistent with the test results, and the model was valid. Based on the formulation of wheat flour/sprouted buckwheat flour/black rice flour/purple sweet potato flour in a ratio of 34.5:28.8:26.7:10.0, compounding and extruding were performed to obtain reconstituted rice; this was further subjected to microwave puffing to obtain instant reconstituted rice. Figure 2 shows the physical drawings of the four raw material flours, namely rice (commercially available), instant rice (commercially available), reconstituted rice, and instant reconstituted rice. The reconstituted rice was purple-black in color, with a compact structure, uniform size, and a cereal aroma. Instant reconstituted rice was brownish-red in color, fully cooked internally, and ready to eat, with a crispy texture and rich grain aroma.

### 3.2. Nutritional Composition, Total Flavonoids, and Total Phenol Content

A comparison of the nutrient composition of the four raw materials and the four rice products is shown in Figure 3a–c. The total starch content of reconstituted rice and instant reconstituted rice was significantly lower than that of the raw materials, rice, and instant rice; this finding was related to starch degradation during the high-temperature extrusion process, wherein some of the starch might be degraded to glucose, maltose, etc. [21]. The protein, fat, and ash content in reconstituted rice and instant reconstituted rice were significantly higher than that in wheat flour, rice, and instant rice. The moisture content in instant rice, reconstituted rice, and instant reconstituted rice was significantly lower than that in the four raw materials and slightly lower than that in rice; this was presumably due to thermal evaporation of internal moisture under the puffing conditions of high temperature and extrusion [22]. Hence, reconstituted rice and instant reconstituted rice are stored longer. During high-temperature extrusion, the macromolecules (mainly starch and proteins) in the raw materials gradually stretch, undergo partial depolymerization, and are degraded into a variety of small molecules, with an increase in the number of polar groups [23] and an enhanced ability to bind with water. This significantly increased the water-soluble content of reconstituted rice and recombinant instant reconstituted rice (approximately 2.7-fold for larger rice). In conclusion, compared to rice and instant rice, reconstituted rice and instant reconstituted rice had a combination of multiple nutrients from the four raw materials, had higher protein and fat contents, and were more resistant to storage due to their lower moisture content.

As depicted in Figure 3d, the levels of total flavonoids and total phenols in reconstituted rice and instant reconstituted rice were lower than those in sprouted buckwheat but higher than those in wheat flour, black rice flour, and purple sweet potato flour. The total flavonoids and total phenols in reconstituted rice were notably higher than those in instant reconstituted rice, suggesting that extrusion and microwave expansion resulted in a reduction in flavonoids and total phenols from the raw materials. This phenomenon can be attributed to the thermal instability of flavonoids and polyphenols, leading to increased decomposition rates during the processes of extrusion and microwave puffing as temperature and shear increase [24,25]. Sneh Punia Bangar et al. [26] reported that the contents of total flavonoids and total phenols of barley flour under extruded puffing conditions were reduced by 16.4–34.2% and 23.4–38.1%, respectively. Additionally, total flavonoids and total phenols were not detected in rice and instant rice. In a like-for-like comparison in the present study, the total flavonoid content (9.98 ± 0.22 mg/g) and total phenol content (10.87 ± 0.10 mg/g) in reconstituted rice and the total flavonoid content (8.15 ± 0.82 mg/g) in instant reconstituted rice were slightly higher than the total flavonoid content (5.2 ± 0.1 mg/g) and total phenol content (8.9 ± 0.2 mg/g) in sprouted buckwheat biscuits prepared by Romina Molinari et al. [27]. In conclusion, because reconstituted rice and instant reconstituted rice retain certain amounts of flavonoids and polyphenols after extrusion and puffing, the health benefits of reconstituted rice and instant reconstituted rice are better than those of rice and instant rice.

### 3.3. In Vitro Digestive Characterization and Dietary GI Analysis

The correlation between the human GI and the hydrolysis index is expressed as follows: GI = 39.71 + 0.549HI. Therefore, the in vitro digestion test can be utilized to initially assess alterations in the GI of the human digestive system’s post-food digestion. In Figure 4a, the glucose release profiles of the four raw materials and the four rice products digested for 3 h are illustrated. The inclines of the curves within the initial 30 min were steeper, suggesting rapid changes in GI values, and the curves gradually leveled off after 60 min. This indicates the completion of in vitro starch digestion, with glucose release values of the four raw materials and rice products being lower than those of the white bread (the reference sample). The rate and amount of glucose release are related to the resistant starch in the sample [28], and a higher percentage of resistant starch content results in a slower rate of glucose release and a smaller amount of glucose release. Extrusion and microwave puffing can increase the content of resistant starch [29,30]; this explains the digestion patterns of the four raw materials and the four rice products presented in Figure 4a.

Dietary GI is a measure of the effect of carbohydrates in food on blood glucose concentrations and is divided into three categories: low-GI foods (GI < 55), medium-GI foods (56 < GI < 69), and high-GI foods (GI > 70) [31]. As shown in Figure 4b, instant rice had the highest GI value (85.51), followed by rice and wheat flour, all of which were high-GI foods. The GI values of instant reconstituted rice, reconstituted rice, purple sweet potato flour, and black rice flour were 69.47, 68.86, 68.67, and 68.41, respectively; these values were in the range of 55 < GI < 70 and belonged to the moderate-GI foods. Reconstituted rice can slow down the increase in the blood glucose level to some extent when compared with rice and instant rice.

### 3.4. Analysis of Color and XRD

The results of chromaticity measurements are depicted in Figure 5a. The brightness values (L*) of reconstituted rice flour and instant reconstituted rice flour were notably higher than those of purple sweet potato flour but notably lower than those of rice flour, instant rice flour, wheat flour, and sprouted buckwheat flour. The reddish-green coloration values (a*) of reconstituted rice flour and instant reconstituted rice flour were significantly lower than those of purple sweet potato flour but significantly higher than those of wheat flour, black rice flour, rice flour, instant rice flour, and sprouted buckwheat flour. The yellow-blue color values (b*) of reconstituted rice flour and instant reconstituted rice flour were notably lower than those of wheat flour and sprouted buckwheat flour but notably higher than those of rice flour, purple sweet potato flour, and black rice flour. The total color difference values (ΔE*) of reconstituted rice flour and instant reconstituted rice flour were significantly lower than those of purple sweet potato flour but significantly higher than those of sprouted buckwheat flour, wheat flour, instant rice flour, and rice flour. Both a* and b* values of instant reconstituted rice flour were significantly higher than those of reconstituted rice flour; thus, a significant browning reaction is indicated during microwave puffing, which is related to the occurrence of a Maillard reaction [32].

Figure 5b shows the XRD patterns of the raw materials and rice products. The information of XRD patterns corresponds to the crystal types of starch, such as A, B, C, and D [33]. As observed in the spectra, sprouted buckwheat, wheat, black rice, purple sweet potato, and rice displayed prominent diffraction peaks around 2θ of 15°, 17°, 18°, and 23°, with continuous double peaks evident near 17° and 18°, indicative of a typical A-type structure. Conversely, fewer diffraction peaks were observed for instant rice, reconstituted rice, and instant reconstituted rice, with changes in peak locations. These samples exhibited stronger diffraction peaks solely near 2θ of 13° and 20°, forming a V-shaped structure. Following extrusion puffing and microwave puffing treatment, the starch in reconstituted rice and instant reconstituted rice changed from the original A-type structure to a V-type structure; this change is associated with the formation of amylose-lipid complexes [34], and this structure slows down the starch digestion rate [35]. The relative crystallinity values of instant rice, reconstituted rice, and instant reconstituted rice were 4.16%, 3.01%, and 2.60%, respectively; these values were considerably lower than those of the four raw materials and rice. This indicates that the cooking, extrusion, and microwave processes disrupted the original starch crystal structure. Simultaneously, amylose and lipids combined to form new complexes, which damaged the crystal zone and increased the amorphous structure. Consequently, the energy required for starch gelatinization was reduced, leading to a decrease in gelatinization enthalpy.

### 3.5. Microstructure and Texture Analysis

As shown in Figure 6, most of the particles of the four raw material flours were spherical or oval in shape and were encapsulated by flaky or lumpy substances, which are probably polysaccharides, proteins, lipids, and other components; this finding is consistent with the reports on the study of wholegrain flour [8,36]. The microstructures of rice, instant rice, reconstituted rice, and instant reconstituted rice showed irregular lumps as compared to those of the four raw material flours. The internal texture of rice was homogeneous, with a tight and well-connected structure. A small number of holes appeared on the surface of instant rice, reconstituted rice, and instant reconstituted rice; the surface holes of reconstituted rice and instant reconstituted rice were relatively larger, with traces of extrusion fracture. This was due to the moisture evaporation and expansion of the gas of the material following extrusion puffing; the structure of the cooled material was altered, and the higher the degree of pasting, the more apparent the appearance of holes was [37]. Under the effect of extrusion, the dietary fiber present in the raw material can cause the internal structure of reconstituted rice to become loose and irregular [38]. Additionally, the surface pore size can directly affect the cooking rehydration rate; the larger the pore size of the product, the faster the rehydration rate, and the shorter the cooking time.

Hardness is the pressure required to allow the food to undergo a certain extent of deformation. Tackiness is the force required to peel off the food and other objects (by using tongue, teeth, or mouth) when simulating their attachment. Springiness refers to the ability of the food to return to its original state after subjecting to a force. Cohesiveness represents the mechanical strength of the adhesion within the structure of the food [39]. Gumminess refers to the energy needed for food to attain a stable state suitable for swallowing, while chewiness indicates the energy required for chewing food into a stable state for swallowing. According to Table 3, no significant disparities were observed in tackiness, springiness, and cohesiveness among rice, instant rice, and reconstituted rice. This suggests that reconstituted rice shares similarities with rice and instant rice regarding stickiness, softness, and mechanical bonding strength. However, the hardness, gumminess, and chewiness of reconstituted rice were notably lower than those of rice. This is likely due to surface cracks on reconstituted rice, facilitating water penetration and decomposition during cooking. Hence, the cooking duration for reconstituted rice can be appropriately reduced while maintaining its palatability.

### 3.6. Analysis of Thermal and Pasting Properties

The paste transition temperature indicates the density of starch crystallization, and the enthalpy is associated with the size of crystallinity [40]. As shown in Table 4, the four raw materials as well as rice, reconstituted rice, and instant reconstituted rice exhibited heat absorption at 40–90 °C due to the water absorption and pasting of starch. The pasting transition temperature of instant rice was significantly higher than that of rice, probably because of the increase in amylose content during the cooking process; amylose forms complexes with lipids, which is not favorable for starch pasting and increases the pasting temperature [41]. The pasting transition temperatures of reconstituted rice and instant reconstituted rice were less than those of the four raw materials. This might be because the degree of pasting of reconstituted rice and instant reconstituted rice became higher after extrusion [42]. Caloric changes were more stable under the heating condition, and the energy required for pasting was less. The enthalpy values of pasting of instant rice, reconstituted rice, and instant reconstituted rice were less than those of the four raw materials and rice; this finding suggests that cooking, extrusion, and microwave treatments can reduce the crystallinity of starch, and the results were consistent with the XRD test results.

Starch gelatinization is an irreversible order-to-disorder transition that occurs during thermal and nonthermal treatments when water is absorbed, and the product is swollen due to excess water [43]. Table 5 shows the pasting properties of the four raw materials and the four rice products. No pasting temperature was detected for instant rice, reconstituted rice, and instant reconstituted rice; this was probably because these three rice products were already pasted during the puffing process. The peak viscosity, holding strength, and final viscosity of instant rice, reconstituted rice, and instant reconstituted rice were notably lower than those of rice and the four raw materials. Viscosity value reflects the extent of starch breakdown. A higher degree of breakdown facilitates easier entry of water molecules into the interior of starch crystals. Moreover, with increasing temperature, micro-crystalline bundles disperse more extensively, rendering them essentially colloidal and close to a molecular state, consequently reducing viscosity [44]. The breakdown value indicates the degree of starch granule rupture and starch stability during heating. A larger breakdown value signifies greater instability in the starch structure [41]. The setback value denotes the anti-retrogradation ability and gelatinization of starch. The smaller the setback value, the better is the anti-retrogradation ability; however, it cannot easily to form gel [45]. The breakdown and setback values of instant rice, reconstituted rice, and instant reconstituted rice were significantly smaller than those of rice; this result indicates that the pasteurization stability of the puffed samples was better, and they could not be easily hardened and undergo retrogradation during cooling. This facilitates the maintenance of rice quality.

### 3.7. Analysis of Volatile Components in Rice Products

To compare flavor differences between reconstituted rice, instant reconstituted rice, commercially available rice, and instant rice, the four rice products were analyzed for flavor by GC-MS. A total of 68 substances were detected by GC-MS (Appendix A), including 16 hydrocarbons, 14 aldehydes, 11 naphthalenes, 5 esters, 4 alcohols, 3 ketones, 3 acids, 2 benzenes, 2 indenes, and 8 other compounds. Totals of 8, 11, 47, and 42 flavor components were detected in rice, instant rice, reconstituted rice, and instant reconstituted rice, respectively. In reconstituted rice, alkanes had the highest content of 172.5 µg/g, followed by esters, naphthalenes, alcohols, acids, and aldehydes. Following microwave puffing of the reconstituted rice to produce instant reconstituted rice, 21 species vanished, while 16 new species emerged in the flavor components. Stacked histograms depicting the number of volatile compound species, volatile compound content, volatile compound percentage content, and a heat map of volatile compound content were plotted to visually analyze the disparities among the four rice products (Figure 7). Although alkanes show a high content in rice, reconstituted rice, and instant reconstituted rice, they are not the main aroma presenters because of their high odor threshold. Generally, typical aromatic volatiles include oxygenated groups, nitrogen groups, sulfur groups, and aromatic groups [46]. Among the rice volatile compound species, cedrol (oxygenated groups) and *N*-methylphenylethanolamine (oxygenated groups and nitrogen groups) were high at 24.37 µg/g and 7.84 µg/g, respectively; this imparted a woody and pungent aroma to the rice. Aldehydes are mainly produced through lipid oxidation and catabolism and contribute the most to the overall flavor because of relatively low odor thresholds [46]. The volatile compound species in instant rice showed a high content of nonanal and 2,4-nonadienal, with nonanal imparting an oily and sweet orange aroma and 2,4-nonadienal imparting a floral, fruity, and oily aroma to instant rice. Volatiles such as 2,4-nonadienal, n-octanal, and (E)-2-octenal were present in both reconstituted rice and instant reconstituted rice. The higher content of (E)-2-octenal renders a fatty and meaty aroma to the product. 2,4-Nonadienal imparts a strong floral, fruity, and oily aroma to the product. n-Octanal gives the product rose and orange-peel-like aromas. Naphthalenes are found only in reconstituted rice and instant reconstituted rice, and they mainly present a pungent odor [47]. Some studies have reported that alcohols, which are byproducts of the oxidation of unsaturated fatty acids, are formed from further decomposition of aldehydes and are the second most abundant volatile substance in rice [48]. However, in our present study, alcohols were found only in rice and reconstituted rice. α-Pinitol in reconstituted rice rendered it a characteristic clove aroma. Additionally, reconstituted rice and instant reconstituted rice contain characteristic flavor substances such as D-limonene (citrus aroma), terpinene (lemon aroma), and cinene (lemon aroma). Heterocyclic compounds such as pyrroles, furans, and pyrazines are the main volatile flavor compounds produced by the Maillard reaction. 2-Pentylfuran was identified in instant rice, reconstituted rice, and instant reconstituted rice, likely resulting from the Maillard reaction during cooking, extrusion puffing, and microwave puffing. At low concentrations, 2-Pentylfuran exhibits a characteristic nutty scent, while at higher concentrations, it presents a less pleasant aroma reminiscent of soybeans. Instant reconstituted rice contains 2-methylpyrazine, imparting a beef-like and nutty aroma. In summary, reconstituted rice and instant reconstituted rice boast a wider array of flavor components and a higher concentration of flavor substances compared to commercially available rice and instant rice.

## 4. Conclusions

In this study, a D-optimal mix design was employed to optimize the blending ratio of sprouted buckwheat flour, black rice flour, wheat flour, and purple sweet potato flour, utilizing response values including α-amylase inhibition, resistant starch content, sensory scores, and weighted scores of the raw materials. Response surface analysis was conducted to examine the interactions among the components of the raw materials. The findings revealed that the content of buckwheat significantly influenced resistant starch content. α-Amylase inhibition exhibited a positive correlation with wheat content, sensory scores were positively associated with black rice content, and the combination of three ingredients demonstrated a synergistic effect. Combined with the optimal solution function of the software, the optimal formula was determined as 34.5% wheat, 28.8% sprouted buckwheat, 26.7% black rice, and 10% purple sweet potato, and the weighted score value was 52.65; the experimental results agreed with the predicted values. Reconstituted rice was then prepared by extrusion following the optimal ratios and showed a purple-black color with a cereal flavor. The reconstituted rice underwent additional microwave puffing to produce instant reconstituted rice, characterized by a reddish-brown color and a cereal flavor. Both reconstituted rice and instant reconstituted rice exhibited a modified starch structure (V-shaped) compared to the raw materials, displaying reduced relative crystallinity, improved thermal stability, and a porous surface microstructure. In comparison to commercially available rice and instant rice, reconstituted rice and instant reconstituted rice contained relatively higher protein and fat contents, lower water content, and functional ingredients like phenols and flavonoids. A total of 47 and 42 flavor components were detected in reconstituted rice and instant reconstituted rice, respectively. Additionally, the hardness, gumminess, and chewiness of reconstituted rice were significantly lower than those of rice. The GI values of reconstituted rice and instant reconstituted rice were 68.97 and 69.47, respectively; these values suggest that reconstituted rice and instant reconstituted rice are moderate-GI foods. Thus, the investigated reconstituted rice and instant reconstituted rice in our study show a unique flavor, are rich in nutrients and functional components, and can alleviate the increase in blood glucose levels to a certain extent, which could benefit people with high blood glucose levels. Instant reconstituted rice can also be developed as a casual snack dish or a nutritious breakfast cereal, which is promising for industrial application.

## Figures and Tables

**Figure 1 foods-13-01148-f001:**
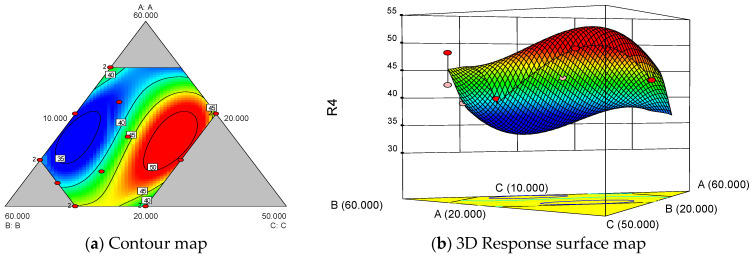
Effects of different ratios of raw material on weighted scores.

**Figure 2 foods-13-01148-f002:**
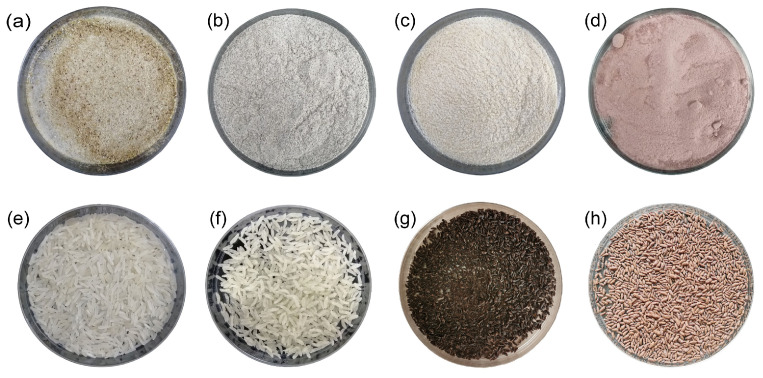
Images of raw materials and rice products ((**a**) sprouted buckwheat flour; (**b**) black rice flour; (**c**) wheat flour; (**d**) purple potato flour; (**e**) rice; (**f**) instant rice; (**g**) reconstituted rice; (**h**) instant reconstituted rice).

**Figure 3 foods-13-01148-f003:**
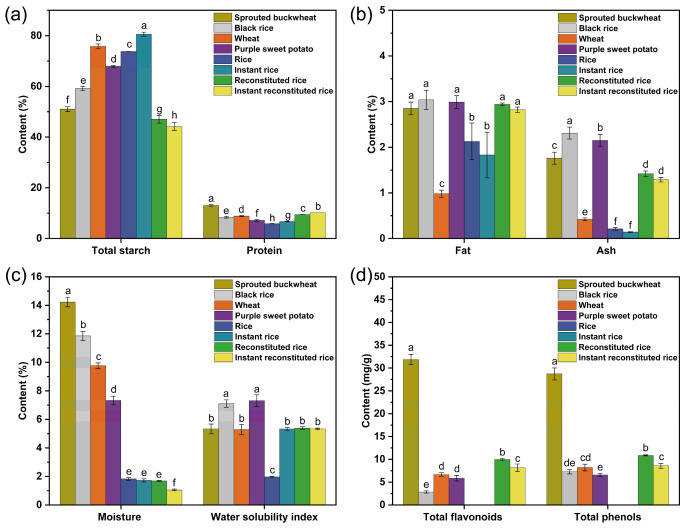
Total starch and protein content (**a**), fat and ash content (**b**), moisture and water solubility index (**c**), and total flavonoid and phenol content (**d**) in raw materials and rice products. Note: different letters indicate significant differences (*p* < 0.05).

**Figure 4 foods-13-01148-f004:**
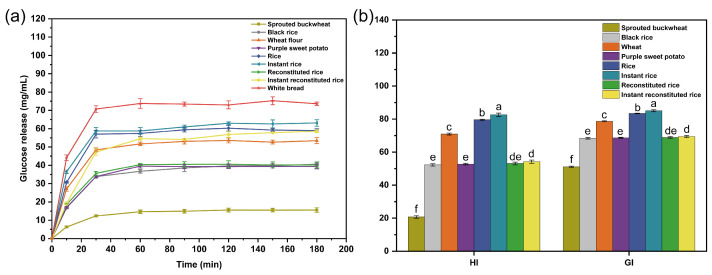
Glucose release curves (**a**) and glycemic index (**b**) of raw materials and rice products. Note: different letters indicate significant differences (*p* < 0.05).

**Figure 5 foods-13-01148-f005:**
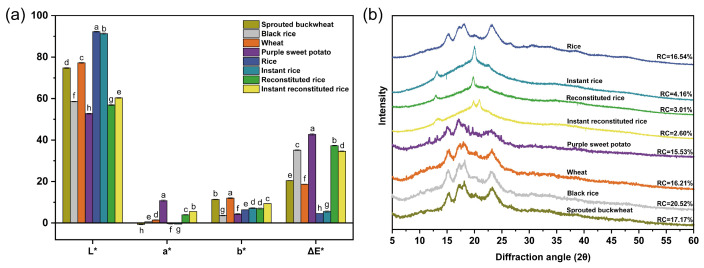
Color (**a**) and XRD spectra (**b**) of raw materials and rice products. Note: different letters indicate significant differences (*p* < 0.05).

**Figure 6 foods-13-01148-f006:**
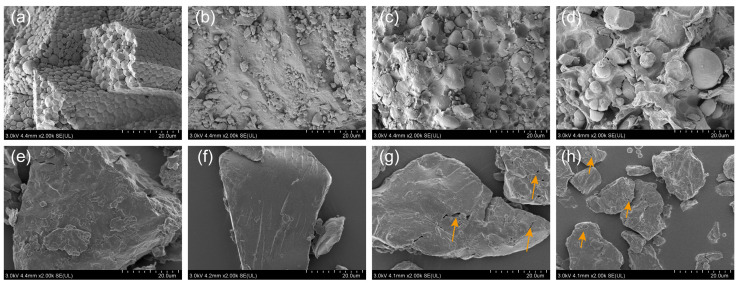
Microstructure of sprouted buckwheat (**a**), black rice (**b**), wheat (**c**), purple sweet potato (**d**), and rice (**e**), instant rice (**f**), reconstituted rice (**g**), and instant reconstituted rice (**h**).

**Figure 7 foods-13-01148-f007:**
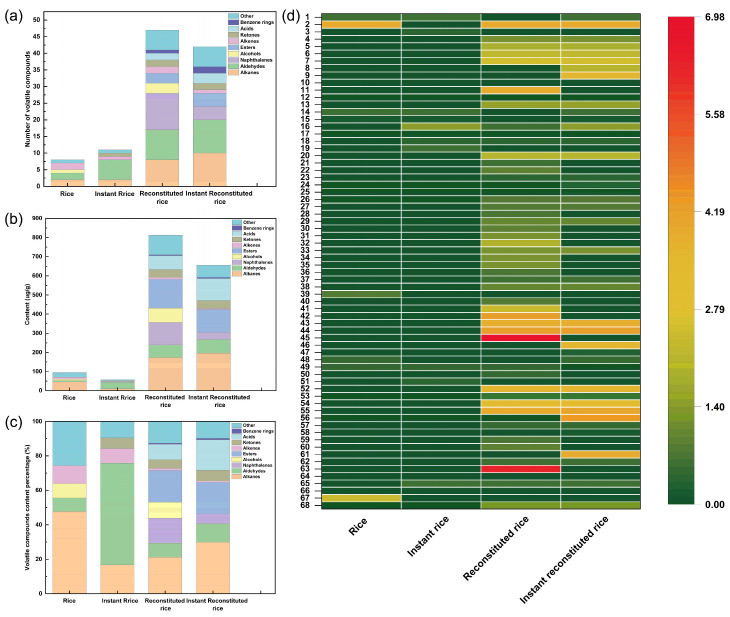
Number of volatile compounds (**a**), volatile compound content (**b**), volatile compound content percentage (**c**), and heatmap (**d**) of volatile compounds of rice, instant rice, reconstituted rice, and instant reconstituted rice. (Note: 1. Decane; 2. Eicosane; 3. Methylcyclooctane; 4. Dodecane; 5. n-Tetradecane; 6. n-Pentadecane; 7. n-Hexadecane; 8. Cyclopentadecan; 9. Nonadecane; 10. Undecane; 11. Phytane; 12. Cycloheptane; 13. n-Tridecane; 14. Nonanal; 15. Hexanal; 16. 2,4-Nonadienal; 17. (E)-Hept-2-enal; 18. n-Octanal; 19. trans-2-Nonenal; 20. (E)-2-Octenal; 21. (Z)-2-Nonenal; 22. Decanal; 23. 2-trans-Decenal; 24. Heptaldehyde; 25. Phenylacetaldehyde; 26. 2,4-Decadienal; 27. trans-2,4-Decadienal; 28. 2-Methyl-1,2,3,4-Tetrahydronaphthalene; 29. 1,5-Dimethyl-1,2,3,4-tetrahydro-naphthalin; 30. 2,6-Dimethylnaphthalene; 31. 1,5,7-Trimethyl-1,2,3,4-tetrahydronaphthalene; 32. 1,2,3,4-Tetrahydro-4-isopropyl-1,6-dimethylnaphthalene; 33. 2,3,5-Trimethylnaphthylene; 34. 6-Ethyl-1,2,3,4-tetrahydronaphthalene; 35. 2,5,8-Trimethyl-1,2,3,4-tetrahydronaphthalene; 36. 5-Methyltetralin; 37. 2-Methylnaphthalene; 38. 1,4-dimethyl-1,2,3,4-tetrahydronaphthalene; 39. N-methylphenylethanolamine; 40. α-Pinitol; 41. Spathulenol; 42. Isophytol; 43. Dibutyl phthalate; 44. Methyl linoleate; 45. l-Ascorbyl dipalmitate; 46. Methyl hexadecanoate; 47. Methyl oleate; 48. Cinene; 49. D-Limonene; 50. Terpinene; 51. 3-Octen-2-one; 52. 6,10,14-Trimethyl-2-pentadecanone; 53. 3-Nonen-2-one; 54. Palmitic acid; 55. Stearic acid; 56. Oleic acid; 57. 1-Isopropyl-2-methylbenzene; 58. o-Xylene; 59. 1,2-Dimethyl-2,3-dihydro-1H-indene; 60. 1,3,3-Trimethyl-1,2-dihydro-indene; 61. Dichloromethane; 62. 1-(Phenylsulfonyl)Pyrrole; 63. Di-tert-dodecyl disulfide; 64. Vinyl isopropyl ether; 65. 2-Pentylfuran; 66. 2-Methylpyrazine; 67. Cedrol; 68. 4,6,8-Trimethyl azulene.

**Table 1 foods-13-01148-t001:** Mixing test design and results.

Test Set	A Sprouted Buckwheat Flour	B Wheat Flour	C Black Rice Flour	D Purple Sweet Potato Flour	R_4_ Weighted Score
1	20.000	50.000	20.000	10.000	42.358
2	30.000	30.000	30.000	10.000	48.676
3	40.000	20.000	30.000	10.000	43.200
4	27.500	42.500	20.000	10.000	41.874
5	50.000	30.000	10.000	10.000	43.484
6	30.000	50.000	10.000	10.000	37.222
7	20.000	40.000	30.000	10.000	38.478
8	35.000	35.000	20.000	10.000	43.636
9	42.500	32.500	15.000	10.000	37.373
10	20.000	50.000	20.000	10.000	48.007
11	40.000	20.000	30.000	10.000	43.128
12	25.000	50.000	15.000	10.000	39.020
13	50.000	30.000	10.000	10.000	44.018
14	20.000	40.000	30.000	10.000	40.011
15	30.000	50.000	10.000	10.000	41.305
16	40.000	40.000	10.000	10.000	36.445

**Table 2 foods-13-01148-t002:** Results of ANOVA for weighted score.

Source	Sum of Squares	df	Mean Square	F-Ratio	Prob > F
Model	166.78	9	18.53	4.34	0.044 *
linear mixed model	24.61	2	12.31	2.88	0.1327
AB	20.78	1	20.78	4.87	0.0695
AC	22.87	1	22.87	5.36	0.0599
BC	19.74	1	19.74	4.62	0.0751
ABC	12.06	1	12.06	2.82	0.1438
AB (A–B)	9.85	1	9.85	2.31	0.1797
AC (A–C)	44.81	1	44.81	10.5	0.0177 *
BC (B–C)	24.46	1	24.46	5.73	0.0538
Residual	25.62	6	4.27		
Lack of fit	0.010	1	0.010	0.002	0.9665
Pure error	25.61	5	5.12		
R^2^ = 0.8668			R^2^_Adj_ = 0.6671		

Note: *p* > 0.05 is not significant; *: *p* < 0.05 is significant.

**Table 3 foods-13-01148-t003:** Textural properties of rice products.

Samples	Hardness (g)	Tackiness (g)	Springiness (mm)	Cohesiveness	Gumminess (g)	Chewiness (mJ)
Rice	57.17 ± 3.18 a	9.33 ± 1.86 a	1.20 ± 0.05 a	1.13 ± 0.05 a	62.05 ± 3.95 a	0.67 ± 0.09 a
Instant rice	23.33 ± 1.20 b	5.83 ± 0.17 a	0.63 ± 0.23 a	1.07 ± 0.17 a	30.73 ± 1.22 b	0.18 ± 0.08 b
Reconstituted rice	18.17 ± 0.17 b	7.33 ± 0.44 a	0.84 ± 0.21 a	1.38 ± 0.32 a	27.93 ± 4.83 b	0.23 ± 0.11 b

Note: different lowercase letters in the same column indicate significant differences (*p* < 0.05).

**Table 4 foods-13-01148-t004:** Thermal properties of raw materials and rice products.

Samples	*T_o_*/°C	*T_p_*/°C	*T_c_*/°C	*ΔH*/g
Sprouted buckwheat flour	60.54 ± 0.67 cd	67.56 ± 0.29 c	78.17 ± 0.86 c	5.76 ± 0.65 a
Black rice flour	60.92 ± 0.31 c	66.67 ± 0.22 d	78.52 ± 1.21 c	6.29 ± 0.37 a
Wheat flour	59.23 ± 0.16 d	64.14 ± 0.85 f	69.21 ± 0.41 ef	4.66 ± 0.22 b
Purple sweet potato flour	65.27 ± 0.84 b	77.27 ± 0.27 a	85.72 ± 1.77 b	4.16 ± 0.47 b
Rice flour	59.63 ± 1.68 cd	67.99 ± 0.35 c	70.54 ± 1.13 e	2.01 ± 0.21 c
Instant rice flour	69.82 ± 0.11 a	76.28 ± 0.21 b	89.13 ± 0.73 a	1.58 ± 0.08 cd
Reconstituted rice flour	54.67 ± 0.61 e	65.19 ± 0.56 e	72.39 ± 1.03 d	1.11 ± 0.12 de
Instant reconstituted rice flour	40.75 ± 0.40 f	50.47 ± 0.06 g	68.56 ± 0.49 f	0.71 ± 0.09 e

Note: *T_o_*, onset temperature; *T_p_*, peak temperature; *T_c_*, conclusion temperature; *ΔH*, gelatinization enthalpy.

**Table 5 foods-13-01148-t005:** Pasting properties of raw materials and rice products.

Samples	Pasting Temperature	Peak Time	Peak Viscosity	Holding Strength	Final Viscosity	Breakdown	Setback
Sprouted buckwheat flour	77.20 ± 0.05 e	6.86 ± 0.13 a	725.67 ± 8.50 d	677.0 ± 15.87 c	1185.0 ± 38.35 d	45.33 ± 4.51 d	510.33 ± 39.55 d
Black rice flour	91.62 ± 0.51 b	6.24 ± 0.14 b	1744.33 ± 129.28 b	1194.67 ± 41.40 b	2783.33 ± 224.26 b	549.67 ± 91.13 b	1588.67 ± 184.94 a
Wheat flour	92.40 ± 0.26 a	5.69 ± 0.08 c	864.33 ± 34.43 c	642.00 ± 32.23 c	1516.33 ± 19.43 c	202.33 ± 31.21 c	854.00 ± 15.10 c
Purple sweet potato flour	84.85 ± 0.05 c	5.20 ± 0.07 d	398.67 ± 10.97 e	333.33 ± 2.08 d	534.67 ± 2.08 e	65.33 ± 9.29 d	201.33 ± 2.08 e
Rice flour	81.92 ± 0.21 d	5.66 ± 0.02 c	3825.50 ± 34.50 a	1881.00 ± 58.00 a	3022.00 ± 56.00 a	1945.00 ± 24.00 a	1141.50 ± 1.50 b
Instant rice flour	-	5.86 ± 0.02 c	377.50 ± 6.50 e	327.50 ± 3.50 d	327.50 ± 3.50 f	50.00 ± 4.00 d	−0.50 ± 0.50 f
Reconstituted rice flour	-	5.64 ± 0.07 c	263.50 ± 6.50 f	219.00 ± 5.00 e	329.50 ± 8.50 f	44.00 ± 2.00 d	110.00 ± 4.00 ef
Instant reconstituted rice flour	-	6.48 ± 0.38 b	82.00 ± 14.00 g	78.00 ± 14.00 f	108.00 ± 11.00 g	3.50 ± 0.50 d	30.00 ± 2.00 f

Note: different lowercase letters in the same column indicate significant differences (*p* < 0.05).

## Data Availability

The original contributions presented in the study are included in the Appendix A, further inquiries can be directed to the corresponding author.

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
