# Peer review of "Effect of Sprouted Buckwheat on Glycemic Index and Quality of Reconstituted Rice"

_foods, 2024, doi:10.3390/foods13081148_

Round 1
Reviewer 1 Report
Comments and Suggestions for Authors
Overall English is poor
Introduction should be rephrased
Line 27: The Buckwheat is a class of food and medicine food crops. Please define buckwheat in a better scientific way, it is not a class of food crops
Line 26 Useful qualities ? define
Line 37 …..the unpalatable nature of buckwheat flour processing, among other factors, restrict the use of buckwheat in functional foods – sentence ?
Eating quality – sensoring quality
Line 85 Buckwheat variety is Jin buckwheat No. 2, origin is Hunan Xiangxi. Sprouted buck- 85 wheat flour laboratory homemade. Wheat flour, black rice flour, purple sweet potato flour, 86 rice, and instant rice are purchased in the market. GIVEN THAT there might be an impact of the wheat flour and other ingredients used (whole flour, quality of flour ….) the way the materials are described here is not correct – needs to be improved – POOR English !
Line 376 – is this in line with literature on extrudates ?
One of the characteristics of the reconstituted rice is the “cereal sensory” – I don’t find which method is used to evaluate the sensory characteristics

Please check the text, some parts should be rephrased. The material and methods is very poorly described. See the high lights in the file.
Reviewer 2 Report
Comments and Suggestions for Authors
Dear corresponding author:
Please address following points to finalize your well written article.
Title
Preparation, characterisation and quality analysis of functional 2 reconstituted rice by sprouted buckwheat
Corresponding Author
[email protected] (G. Y. Li); [email protected](Q.T.Xie)
Summary of Review by Ammar Saleem:
Overall the study is well designed and includes all important experiments to make plausible conclusions regarding the determination of dietary benefits of the modified product - the reconstituted rice.
However, the readers will always be intrigued to know exactly what changes occurred at molecular level during the development of new product. In this regard total phenolics and total flavonoid tests remain ambiguous considering the fact that more specified analytical setups and methods are available. I would recommend that authors address this fact in the discussion.
Title
Ok
Abstract
OK
Key words
Ok
Introduction
L27-30. Add references.
L37-38. Add references.
L61-63. Add reference/s.
Methods
OK
Results
OK
Discussion
See summary of Review.
Conclusions
OK
Supplementary material
Not evaluated
Conflict of Interest
Not evaluated
Author contribution
Not evaluated
Acknowledgements
OK
References
Add references as per suggested.
Reviewer 3 Report
Comments and Suggestions for Authors
Manuscript ID: foods-2920436
Type of manuscript: Article
Title: Preparation, characterisation and quality analysis of functional reconstituted rice by sprouted buckwheat
Title: It must be changed since the functional product is not a functional reconstituted rice by sprouted buckwheat but it contained a lot of ingredients other than rice and buckwheat. The change must be done throughout the manuscript.
Keywords: medium-GI food …it is not a clear keyword…..please, change.
Introduction
Line 27, Buckwheat: a lowercase letter must be used. Furthermore, it is not a class of food
Lines 30-31: Replace ” Not only that, but” with “Moreover” or similar tems
Line 32: replace “etc., and” with “while”
Line 33: replace “, etc., and the” with “. The”
Line 39 and throughout the manuscript: Please, remember that buckwheat is not a cereal
Line 71-71, In this study, the experimental design was carried out by D-Optional in Design 71 Expert 8.0.6.: please, explain what it is.
Line71-80: it is a synthesis of materials and methods and must be removed from the introduction.
Conclusions: They must not contain the summary of the results.
The supplementary materials seem to be another parallel manuscript. It is an unacceptable approach. The manuscript must be understandable as it is and the supplementary materials possibly only need to provide a greater level of analysis.
Consequently, some supplementary material (in a more condensed form) must be included in the original manuscript.
Comments on the Quality of English LanguageThe English is sometimes hard to understand. A complete language revision is necessary.
Round 2
Reviewer 1 Report
Comments and Suggestions for Authors
I appreciate the imrpovements to the document !
Reviewer 3 Report
Comments and Suggestions for Authors
The manuscript has been greatly improved.
Just the last issue: the supplementary material title must be changed accordingly to the new title of the manuscript.